# FLAT-CHAT: A WORD RECOVERY ATTACK ON FEDERATED LANGUAGE MODEL TRAINING

## ABSTRACT

Gradient exchange is widely applied in collaborative training of machine learning models, including Federated Learning. Curious-but-honest participants could potentially infer the output labels in recently used training data by analyzing the latest gradient updates. Previous works mostly demonstrate the attack performance under constraint training settings, such as dozens of short sentences in a batch and a small output space for labels. In this work, we propose a novel gradient flattening attack on the last linear layer of a language model, which significantly improves the attacker's efficiency in inferring the words used in training. We validate the capability of the attack on two language generation tasks: machine translation and language modeling. The attack environment is scaled up to industrial settings of a large output vocabulary and realistic training batch sizes. To mitigate the negative impact of the new attack, we explore two defense methods and demonstrate that adding differential privacy with small noise could effectively defend against our new attack without degrading model utility.

## 1 INTRODUCTION

Distributed collaborative learning has become a prevalent paradigm in training large-scale machine learning models, allowing participants to retain their private datasets in their own silos. Federated Learning (FL), as demonstrated in Figure 1, a server coordinates the training process by collecting the gradient updates from local clients and then synchronizing the global model's parameters to local clients. While the private data of the participants do not require direct sharing with others, the exchanged parameters or gradients of the model remain susceptible to tracking, monitoring, and potential misuse by honest-by-curious participants or malicious attackers, *e.g.,* by reconstructing a batch of data recently used for training (Dang et al., 2021; Zanella-Béguelin et al., 2020).

This work focuses on the data reconstruction threat on federated Large-scale Language Model (LLM) training using exchanged model gradients. Inspired by the sparsity property of gradients from the last linear layer, we apply a matrix flattening operation on the gradient matrix. Then, we derive a simple but effective word reconstruction attack on the flattened gradient vector based on *i)* a theory that its element values follow a two-cluster Gaussian Mixture Model and *ii)* three remarks on the statistical properties of the distribution. Inspired by the theoretical analysis, we design a fast and effective gradient flattening attack FLATCHAT. Our empirical study shows that the new attack provides real-time inference results on large-scale training settings such as a training batch using totally more than 100 thousand token instances (involving over three thousand unique word types) and language models with more than 50 thousand candidate words in their output vocabulary. Compared with the existing attack on the last layer (Dang et al., 2021), our attack is far more effective as the inference time is reduced to a linear order of the matrix size by the last linear layer (several seconds on a single CPU server) versus their complaint of high order time complexity on the matrix size.

Our attack exposes the risk of privacy leakage in collaboratively training language models under a setting that intuitively can be seen as one that improves privacy: large training batch and large vocabulary size. To mitigate the risk of attack, we evaluate two defense methods, freezing the gradients (used by Gupta et al. (2022)) and Differentially Private Stochastic Gradient Descent (Abadi et al., 2016), against our attack. The former method is a strong defense method but it sacrifices the models' performance. The latter manages to mitigate our attacks while obtaining improved model performance.

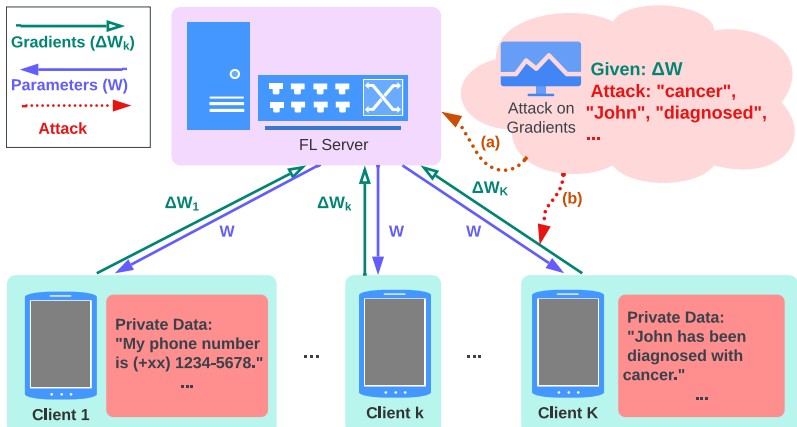

Figure 1: The workflow of Federated Learning (FL) on Language Model, where *i)* the clients upload gradients $\Delta \boldsymbol{W}$ to server (green) and *ii)* server synchronizes the latest model parameters $\boldsymbol{W}$ to clients (blue). The gradient attack (red) accesses $\Delta \boldsymbol{W}$ by (a) a curious-but-honest server, or (b) hackers who monitor the communication channel between them. Then, the attackers infer the private data such as used tokens by a client, given the model gradients $\Delta \boldsymbol{W}$. The figure is best viewed in color.

## 2 RELATED WORK

Private data used in training are shown to be leaked by analyzing the gradients shared in distributed machine learning (Melis et al., 2019; Boenisch et al., 2021), federated learning (FL) as a prevalent sample (Yang et al., 2019; Kairouz et al., 2021). Prior work demonstrated the possibility of reconstructing training images from gradients by optimizing the similarity between the retrieved gradients and the gradients calculated by some heuristic data (Zhu et al., 2019; Zhao et al., 2020). Then, the data reconstruction attack is extended to a more challenging scenario of inferring training sentences, namely sequences of tokens (Zhu et al., 2019; Deng et al., 2021; Gupta et al., 2022; Balunovic et al., 2022). Such attacks are required to provide precise information on token usage and their order, however, they usually work on medium to small batches as summarized in Table 1.a. Label inference attack mainly focuses on inferring output labels of a training sample (Ma et al., 2023; Dang et al., 2021), which are the token instances used in next word prediction when training language models (Radford et al., 2019; Kenton and Toutanova, 2019). The previous work largely relies on matrix factorization and linear programming on the gradient matrix of the last linear layer, which could be extremely slow for LLM as the vocabulary size can easily grow beyond 50 thousand such as GPT-2 (Radford et al., 2019) and even worse for multilingual translation such as NLLB (NLLB et al., 2022) uses a vocabulary of 256,000 tokens. FLATCHAT is a fast and precise label inference attack as it analyzes more concise flattened gradient vectors. Only the attacks on embedding layers achieved similar success on inferring input tokens used in large training batches (Gupta et al., 2022), although attacking embedding has its limitations, *e.g., i)* it can be easily defended by freezing the gradients from the embedding layer; and *ii)* embedding layer corresponds more to the inputs, which can contain different information than outputs.[1] A comprehensive comparison between our work and other label inference attacks with settings and performance is demonstrated in Table 1.b.

## 3 GRADIENT FLATTENING ATTACK FOR INFERRING TRAINING TOKENS

Our attack identifies the tokens used in training via predicting output labels $Y \in \mathcal{Y}$ in a batch of training data $\mathcal{B}$, given the access to the corresponding gradients $\Delta \boldsymbol{W}$ by the last linear layer of a language model.

---

[1]The input and output of a generative model can be different in many tasks, such as machine translation and image captioning. Protecting privacy in both spaces is equally important.

Table 1: An overview of the different attacks on FL of a language model. $\Delta \boldsymbol{W}$ indicates the gradient of the parameters. $b$ and $l$ are the number of sentences and max length of these sentences in a batch. $|\mathcal{T}| = |\{\mathcal{B}\}|$ is the number of word types, the set of words, in $\mathcal{B}$. $D$ is the dimensional size of the hidden representations by the attacked language model. $\mathcal{V}$ is the output vocabulary. $|\mathcal{X}|$ indicates the size of $\mathcal{X}$, *e.g.*, $\{\mathcal{B}\}$, $\mathcal{V}$, and $\Delta \boldsymbol{W}$.

| Attack Method | $\Delta \boldsymbol{W}$ | $\boldsymbol{W}$ | Batch Size | Speed | Task |
|---|---|---|---|---|---|
| DLG (Zhu et al., 2019) | All layers | All layers | $b = 1$ | Medium ($\sim 30$ iterations on LLM) | Inferring input $X$ and output $Y$ |
| LAMP (Balunovic et al., 2022) | All layers | All layers | $b \le 4$ $l \le 27$ | Medium ($\sim 30$ iteration on LLM) | Inferring input sequence $X$ |
| FILM (Gupta et al., 2022) | Embedding | All layers | $b \le 128$ | Medium (Beam search on LM and post-filtering) | Inferring output sequence $Y$ |

(a) Recovering full sentences in a batch.

| | | | | | |
|---|---|---|---|---|---|
| FILM (BoW extraction) (Gupta et al., 2022) | Embedding | N/A | Very Large | Extremely Fast (seconds) | Inferring work types in input $X$ |
| RLG (Dang et al., 2021) | last linear | N/A | Medium ($|\mathcal{T}| \le D$) | Extremely Slow (hours when $|\mathcal{V}| \ge 20K$) | Inferring work types in output $Y$ |
| FLATChat(Ours) | last linear | N/A | Very Large ($|\mathcal{B}| \ge 100K$) | Extremely Fast ($\mathcal{O}(|\Delta \boldsymbol{W}|)$, seconds) | Inferring work types in output $Y$ |

(b) Recovering word types in a batch.

## 3.1 PRELIMINARIES: ATTACKING GRADIENTS OF THE LAST LINEAR LAYER

Here, we sketch the structure of a Large Language Model (LLM) as multiple transformer layers stacked on top of an embedding module and the loss calculation follows:

I. A *backbone model* transforms the input $X$ to a high-dimensional representation, $\boldsymbol{h} = f(X)$;

II. A *linear layer* transforms $\boldsymbol{h}$ into a logit vector with the same size as the vocabulary, $\boldsymbol{z} = \boldsymbol{W}\boldsymbol{h} + \boldsymbol{b}$, where the bias $\boldsymbol{b}$ optional and often omitted in many LLMs;

III. *Softmax* gives the probability of all words: $\boldsymbol{p} = \text{softmax}(\boldsymbol{z})$, where $p_i = \frac{\exp(z_i)}{\sum_j \exp(z_j)}$;

IV. *Cross Entropy Loss* (CE) for training is defined as $\mathcal{L}(\boldsymbol{p}, \boldsymbol{y}) = -\sum_i y_i \log(p_i)$, where $y_i$ is the output labels indicating the tokens used in a training sample $(X, Y)$, where $Y$ is a sequence of tokens $\langle y_1, \cdots, y_i, \cdots, y_{|Y|} \rangle$ and $y_i$ is a token identity in a vocabulary $\mathcal{V}$.

**Threat Model.** In FL, the attackers manage to access the gradients of a language model $\Delta \boldsymbol{W}$ generated by a participant, *e.g.,* an honest-but-curious server receives gradient updates from some clients every epoch. The gradients are in accordance with a batch of data used in recent training,

$$\Delta \boldsymbol{W} \triangleq \sum_{(\boldsymbol{X}_i, \boldsymbol{Y}_i) \in \mathcal{B}} \frac{\partial \mathcal{L}(\boldsymbol{X}_i, \boldsymbol{Y}_i)}{\partial \boldsymbol{W}}, \text{ which is abbreviated as } \sum_i \frac{\partial \mathcal{L}_i}{\partial \boldsymbol{W}}. \tag{1}$$

Our attack algorithm $\mathcal{A}$ works on inferring the set of unique words (word types) $\mathcal{T} \triangleq \{\mathcal{B}\}$ in a training batch $\mathcal{B}$.[2]

## 3.2 FLATTENING ATTACK ON LAST LAYER GRADIENTS

**FL**attening **AT**tack on **Cha**nged gradien**T** (FLATCHAT) is a simple but efficient transformation on gradients matrix $\Delta \boldsymbol{W} \in \mathbb{R}^{|\mathcal{V}| \times D}$ by aggregating its columns to a vector of vocabulary size, $\boldsymbol{s} \triangleq \Delta \boldsymbol{W} \cdot \boldsymbol{1} \in \mathbb{R}^{|\mathcal{V}|}$. We observe that the values in $\boldsymbol{s}$ follow the mixture of two Gaussian distributions comprising word types used or not used in the training batch, which we discuss below. Then, we highlight three remarks, which assist on *i)* deciding which cluster corresponds to the gradient contributions from *'positive'* tokens utilized in a batch (Rmk. 1); *ii)* estimating the number of unique

---

[2]Assuming for causal language models, the input $X$ and output $Y$ are approximately the same, *i.e.,* $X = S + [\text{EOS}]$ and $X = [\text{BOS}] + S$ where $S$ is an original sentence in the corpora. In encoder-decoder architectures, input and output may differ.

---

**Algorithm 1** Gradient Flattening Attack (FLATCHAT).

    **Input:** $\Delta W$ derived from a batch of training samples $\mathcal{B}$.
    **Output:** A set of unique token types $\mathcal{T}$ used in $\mathcal{B}$.
1:   $s \leftarrow \Delta W \cdot \mathbf{1}$             ▷ Gradient flattening (Eqn. 5).
2:   $s \leftarrow \mathrm{Norm}(s)$             ▷ Vector normalization by $s/\|s\|$.
3:   $\boldsymbol{\mu}, \boldsymbol{\sigma}, \boldsymbol{\phi} \leftarrow \mathrm{GaussianMixture}(s, 2)$      ▷ Gaussian Mixture Model with 2 clusters.
4:   $\mu_p, \sigma_p, \phi_p \leftarrow \mathrm{PositiveCluster}(\boldsymbol{\mu}, \boldsymbol{\sigma}, \boldsymbol{\phi})$      ▷ Find the positive cluster (Rmk. 1).
5:   $K \leftarrow \mathrm{PredictCount}(\phi_p)$      ▷ Predict the total number of unique token types (Rmk. 2).
6:   $\mathcal{T} \leftarrow \mathrm{SelectTopK}(s, K, \boldsymbol{\mu}, \boldsymbol{\sigma})$      ▷ Generate a set of K tokens with highest scores (Rmk. 3).
7:   **return** $\mathcal{T}$

---

word types used in the training batch (Rmk. 2); *iii)* ranking the likelihood of tokens being used by contrasting the probability of the token in positive cluster vs. negative cluster (Rmk. 3). Combining our theoretical analysis on the aggregated gradients of the last linear layer, our attack algorithm (FLATCHAT) is demonstrated in Algorithm 1.

**Lemma 1 (Single-Sample Gradient Flattening)** *We start our discussion with a simple case involving only one training sample,* i.e., *one token instance in a batch when training a language model. The row-sum of weight matrix gradients $\Delta W$ derived from a single sample is proportional to the gradients regarding the linear layer's output logits, $g \triangleq \partial\mathcal{L}/\partial z$,*

$$s \triangleq \Delta W \cdot \mathbf{1} = \alpha \cdot g, \tag{2}$$

*where scalar $\alpha$ is defined as the sum of hidden states $\alpha \triangleq \sum_{d=1}^{D} h_d = h^T \cdot \mathbf{1}$ and $D$ is the dimensional size of $h$,* i.e., *$h \in \mathbb{R}^D$.*

**Proof** *Lemma 1 can be proved based on the decomposition of the weight gradients for linear transformation (Geiping et al., 2020) accompanying the definitions of $s$ and $g$.*

$$s = \Delta W \cdot \mathbf{1} = \frac{\partial\mathcal{L}}{\partial W} \cdot \mathbf{1} = \left(\frac{\partial\mathcal{L}}{\partial z} \cdot h^T\right) \cdot \mathbf{1} = \frac{\partial\mathcal{L}}{\partial z} \cdot \left(h^T \cdot \mathbf{1}\right) = \frac{\partial\mathcal{L}}{\partial z} \cdot \alpha = \alpha \cdot g. \tag{3}$$

**Lemma 2 (Multi-sample Gradient Flattening)** *We extend our discussion to a batch with multiple samples,* i.e., *$|\mathcal{B}|$ token instances in a batch $\mathcal{B}$. $\Delta W$ the gradient on $\mathcal{B}$, which is written in the product of two extended matrices, gradient matrix $G$ and hidden representation matrix $H$,*

$$\Delta W \triangleq \sum_{i=1}^{|\mathcal{B}|} \frac{\partial\mathcal{L}_i}{\partial W} = G \cdot H^T, \text{where } H = [-h_i-] \in \mathbb{R}^{D \times |\mathcal{B}|} \text{ and } G = [-g_i-] \in \mathbb{R}^{|\mathcal{V}| \times |\mathcal{B}|}. \tag{4}$$

*The row-sum of the gradient weight matrix is a linear combination of $g_i$,*

$$s \triangleq \Delta W \cdot \mathbf{1} = \sum_{i=1}^{|\mathcal{B}|} \alpha_i \cdot g_i \tag{5}$$

*where $\alpha_i$ is the sum of the $i$-th column $h_i$ in $H$,* i.e., *$\alpha_i = \sum_{d=1}^{D} h_{d,i}$.*

Note that our following discussion focuses on $s$ derived from $\mathcal{B}$ with many training samples.

**Proof** *We note the extension involving multiple word instances by stacking $g$ to $G$ and $h$ to $H$.*

$$s = \Delta W \cdot \mathbf{1} = \left(\sum_i \frac{\partial\mathcal{L}_i}{\partial W}\right) \cdot \mathbf{1} = (G \cdot H^T) \cdot \mathbf{1} = G \cdot (H^T \cdot \mathbf{1}) = G \cdot \boldsymbol{\alpha} = \sum_i \alpha_i \cdot g_i. \tag{6}$$

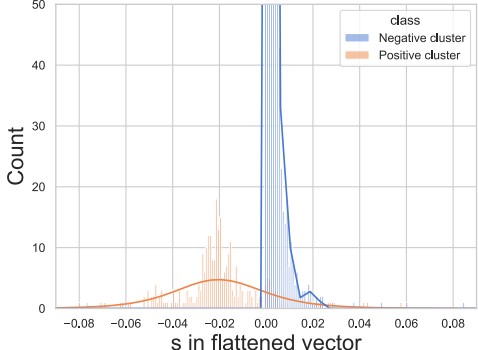 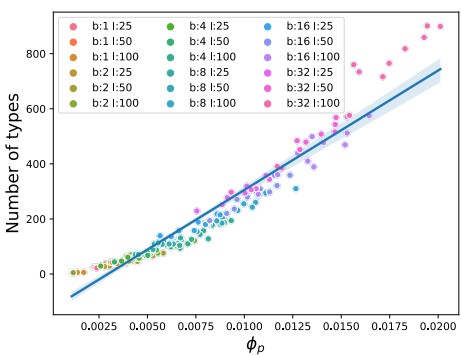

Figure 2: Histogram over values $s$ in the flattened vector $\boldsymbol{s}$ by GPT-2 trained on a $b = 8 \times l = 100$ batch, and the distributions of values positive and negative clusters in two different colors respectively. The number of word types is 299 and the number of token instances is 659 in this batch.

Figure 3: The scatter plot of weights of the positive clusters $\phi_p$ and the numbers of token types $|\mathcal{T}|$ in the same batches, when training GPT-2 on WIKITEXT. We consider various batch shapes $(b \times l)$ and 10 samples for each setting. A linear regression model is plotted with std errors.

**Theorem 3 (Mixture Distribution of Values in $\boldsymbol{s}$)** *The distribution of the values $s$ in $\boldsymbol{s} \in \mathbb{R}^{|\mathcal{V}|}$ is approximated to the mixture of two Gaussian distributions $\mathcal{N}$,*

$$s \sim \phi_p \cdot \mathcal{N}(\mu_p, \sigma_p) + \phi_n \cdot \mathcal{N}(\mu_n, \sigma_n), \text{ where } \phi_p + \phi_n = 1. \tag{7}$$

*The positive ($p$) cluster and negative ($n$) cluster correspond to those word types that have been seen or not have been seen in $\mathcal{B}$, i.e., $t_p \in \mathcal{T}$ and $t_n \in \mathcal{V}/\mathcal{T}$ where $\mathcal{T} = \{\mathcal{B}\}$.*

We illustrate an example histogram on the element values $s$ in the flattened vector $\boldsymbol{s}$ in Figure 2. The values from both positive and negative clusters clearly follow their own distributions with a shape similar to normal distributions and the positive one is much flatter than the negative one. Please find more examples with various batch shapes in Appendix A. The reason for the mixture of Gaussian distribution is discussed as follows.

**Proof** *The value in $\boldsymbol{s}$ with regard to the $t$-th token in dictionary $\mathcal{V}$ could be written as*

$$s_t = \sum_i \alpha_i \cdot g_{t,i} = \sum_{i \text{ s.t. } y_i = t} \alpha_i \cdot g_{t,i} + \sum_{i \text{ s.t. } y_i \neq t} \alpha_i \cdot g_{t,i} \tag{8}$$

$$= \underbrace{\sum_{i \text{ s.t. } y_i = t} \alpha_i \cdot (p_{t,i} - 1)}_{\text{Positive token types.}} + \underbrace{\sum_{i \text{ s.t. } y_i \neq t} \alpha_i \cdot p_{t,i}}_{\text{Negative token types.}} \tag{9}$$

$g_{t,i}$ *is the $t$-th row and $i$-th column of gradient matrix $\boldsymbol{G}$, where $t$ indicates the index of a token in the vocabulary and $i$ means the index of (token-level) training sample in a batch. Similar index notations are used on $p_{t,i}$ with regard to word probability matrix $\boldsymbol{P}$. The gradient in Eqn. 9 is specific to cross-entropy loss on softmax. The two components could be approximated by two weighted Gaussian Distributions, i.e., $\phi_p \cdot \mathcal{N}(\mu_p, \sigma_p)$ and $\phi_n \cdot \mathcal{N}(\mu_n, \sigma_n)$ for each sum in Eqn. 8, where $\phi$ indicates the weights for the clusters. We apply Central Limited Theorem 4 by taking $A_i = \alpha_i$ and $X_i = g_{t,i}$, and considering the numbers of elements in both positive and negative clusters are large, $|\mathcal{B}|$ and $(|\mathcal{V}| - 1) \cdot |\mathcal{B}|$ respectively, given a large training batch with several hundred tokens or more.*

**Theorem 4 (Central Limited Theorem (CLT) on Two Sequences of Variables)** *Say $\{A_i\}$ is a sequence of independent, identically distributed (i.i.d.) random variables which are independent from another sequence of variables, $\{X_i\}$, we define*

$$\overline{S}_n \triangleq \frac{A_1 \cdot X_1 + \cdots + A_n \cdot X_n}{n}.$$

*When $n$ approaches infinity, there exist $\mu_*$ and $\sigma_*$, which assure the random variables $\sqrt{n}(\overline{S}_n - \mu_*)$ almost surely converges to a normal distribution $\mathcal{N}(0, \sigma_*^2)$:*

$$\sqrt{n}(\overline{S}_n - \mu_*) \xrightarrow{a.s.} \mathcal{N}(0, \sigma_*^2) \tag{10}$$

**Proof** *We construct an i.i.d random variables sequence $\{Y_i\}$, where $Y_i = A_i \cdot X_i$. Then, we have the mean $\mu_*$ and variance $\sigma_*$ of $Y$. Theorem 4 can be derived by applying Lindeberg–Lévy CLT (Billingsley, 1961) on the sequence of constructed variables $\{Y_i\}$.*

Given our analysis on the distribution of values in $\boldsymbol{s}$, we propose to use a two-cluster Gaussian Mixture Model (*GMM*) (McLachlan and Basford, 1989) to fit the distribution of $s$. Then, we highlight three important properties of the mixture distribution, which are utilized in designing FLATCHAT.

**Remark 1 (Positive vs Negative Cluster)** *Given a large vocabulary $\mathcal{V}$ in LLM and a training batch $\mathcal{B}$, there are far fewer positive observations, $n_p = |\mathcal{B}|$, than negative observations, $n_n = (|\mathcal{V}|-1)\cdot|B|$. Then, the positive and negative clusters could be identified by $\sigma_p \gg \sigma_n$, as the denominators in Eqn. 10 follow $\sqrt{n_p} \ll \sqrt{n_n}$.*

A series of histogram plots of $s_t$ derived from training GPT-2 on batches with various shapes are illustrated in Appendix A.

**Remark 2 (Correlation between $\phi_p$ and $|\mathcal{T}|$)** *The weight of the distribution regarding the positive cluster $\phi_p$ should be positively correlated to the number of unique words (types) $|\mathcal{T}|$ in a training batch $\mathcal{B}$.*

Intuitively, a larger batch means more positive word types in vocabulary would have been involved in training, meaning a higher weight to the positive cluster. In Figure 3, we observe a clear positive correlation between those two variables. Polynomial regression models with various degrees are trained to estimate $|\mathcal{T}|$. The experiments for comparing those estimators are discussed in Appendix B.

**Remark 3 (Ranking)** *The word types in the positive cluster can be discriminated by the ranking score:*

$$r_t = \left(\frac{s_t - \mu_n}{\sigma_n}\right)^2 - \left(\frac{s_t - \mu_p}{\sigma_p}\right)^2 \tag{11}$$

The discriminative score for deciding the usage of a token is

$$\frac{\mathbb{P}(t \in \mathcal{B}|s_t)}{\mathbb{P}(t \notin \mathcal{B}|s_t)} = \frac{\mathbb{P}(s_t, t \in \mathcal{B})}{\mathbb{P}(s_t, t \notin \mathcal{B})} = \frac{\mathbb{P}(s_t|t \in \mathcal{B})}{\mathbb{P}(s_t|t \notin \mathcal{B})} \cdot \frac{\mathbb{P}(t \in \mathcal{B})}{\mathbb{P}(t \notin \mathcal{B})} \propto \frac{\mathbb{P}(s_t|t \in \mathcal{B})}{\mathbb{P}(s_t|t \notin \mathcal{B})} \tag{12}$$

$$= \frac{\sigma_n \cdot e^{\frac{1}{2}(\frac{s_t - \mu_n}{\sigma_n})^2}}{\sigma_p \cdot e^{\frac{1}{2}(\frac{s_t - \mu_p}{\sigma_p})^2}} \propto \frac{e^{\frac{1}{2}(\frac{s_t - \mu_n}{\sigma_n})^2}}{e^{\frac{1}{2}(\frac{s_t - \mu_p}{\sigma_p})^2}} \propto \frac{1}{2}\left(\left(\frac{s_t - \mu_n}{\sigma_n}\right)^2 - \left(\frac{s_t - \mu_p}{\sigma_p}\right)^2\right). \tag{13}$$

Note that $\mathbb{C}_{\mathcal{B},\mathbb{P}} \triangleq \mathbb{P}(t \in \mathcal{B})/\mathbb{P}(t \notin \mathcal{B})$ is a positive constant value given $\mathcal{B}$ in each flattening attack, therefore $\mathbb{C}_{\mathcal{B},\mathbb{P}}$ has no effect on the ranking. Similarly, $\mathbb{C}_{\mathcal{B},\sigma} \triangleq \sigma_n/\sigma_p$ is also a positive constant value, which does not influence the ranking given the training batch.

## 4 EXPERIMENTS

Previously, we introduced an attack based on flattened gradients by the last linear layer. In this section, we first compare our approach with a previous attack for label inference Dang et al. (2021) in machine translation (MT) experiments. Then, we demonstrate the efficacy of our FLATCHAT in language modeling using increasingly large training batches. An ablation study is provided to compare our scoring metric with a naive word ranking method. Additionally, we discuss the influence of defense methods, such as parameter freezing and differential privacy on parameter gradients.

### 4.1 EXPERIMENTAL SETUP

Our experiments will cover large-scale language models of two applications: *i)* Machine Translation (MT) experiments for comparing our approach against a baseline method RLG, and *ii)* Large Language Modeling (LLM) for validating the performance of FLATCHAT on models with large vocabulary sizes.

**Large Language Modeling.** We initialize our model using a pre-trained GPT-2 language model with a vocabulary size of 50,257 (Radford et al., 2019).[3] We simulate the attack by computing gradients for the model over random batches of text drawn from a holdout dataset WIKITEXT (Merity et al., 2017), of varying sizes and shapes. We consider all combinations of batch sizes of $b \in \{1, 2, 4, 8, 16, 32\}$ sentences and maximum sentence length $l \in \{25, 50, 100\}$, reporting the sizes as $b \times l$. For each setting, we collect 20 gradients using the pre-trained model weights, and train a linear regressor to predict the number of word types[4] in the batch. We report the performance of our attack by training clients on their own for several SGD steps[5] on datasets from different *unseen* domains, *i.e.,* IMDB (Maas et al., 2011) and AGNEWS (Zhang et al., 2015), and with a range of batch sizes including settings unseen in regression training. We report the averaged precision, recall and F-1 score of word type prediction over 10 test batches.

**Machine Translation.** We conduct attack experiments on the IWSLT 2017 de-en corpus (IWSLT). We consider two experimental settings for MT: attacking models trained from scratch (Scratch) or with fine-tuning (FineTune). The latter model was pre-trained on a holdout dataset, NEWSCOMMEN-TARY v15 de-en for 5 epochs, followed by fine-tuning on IWSLT. We implement our experiments using FAIRSEQ (Ott et al., 2019), employing a Transformer (Vaswani et al., 2017) and a byte pair encoding (Sennrich et al., 2016) tokenizer with a vocabulary size of 16,594. We chose a moderate vocabulary size to ensure a fair comparison with the baseline method. Similar to LLM setting, we use 20 randomly selected batches with maximum batch size in $\{25, 50, 100, 250, 500, 1000, 1250, 1500, 2000\}$ from NEWSCOMMENTARY for word type regression. We also report the averaged precision, recall and F-1 score over 10 test batches on IWSLT.

**Attack Methods.** As a baseline, we use Revealing Labels from Gradients (RLG) (Dang et al., 2021) method, which predicts the number of word types based on the matrix rank of $\Delta W$ and predicts those types by identifying separable vectors corresponding to used word types using linear programming (Vaidya, 1989). Note that such an estimation strategy for word type number cannot handle $|\mathcal{T}| > D$ because $\text{rank}(\Delta W) \leq \min\{D, |\mathcal{V}|\}$ is practically upper-bounded by $D$. Regarding the efficiency of the attack, RLG requires solving a linear programming problem with a $D \times |\mathcal{V}|$ parameter matrix $|\mathcal{V}|$ times as the worst case, while FLATCHAT achieves the same order of computing time as the matrix size $\mathcal{O}(|\Delta W|) = \mathcal{O}(|\mathcal{V}| \cdot D)$ and the calculation can be further optimized by using a GPU.[6] For our method, we employ a simple linear regression estimator $\mathcal{R}$ for the number of word types,[7] and compare this against a sky-line ground-truth (ORAC.), which uses the ground truth number of word types. We also report an ablation where we compare the proposed Gaussian Mixture confidence method (*GMM*) for identifying word types against a simpler technique, *Abs.*, which selects types based on the absolute value of their flattened gradient, $|s_t|$. *Abs.* is motivated by the sparsity of $s$, as elaborated in the proof in Appendix C. More details about the attack environment are provided in Appendix D.

## 4.2 RESULTS AND DISCUSSION

### 4.2.1 COMPARISON WITH BASELINE LABEL RECOVERING ATTACK

We compare FLATCHAT with RLG on machine translation experiments in Table 2. Given a medium size batch, say 100 or 500 token instances, FLATCHAT achieves very high inference accuracy on Scratch models and acquires decent attack results on FineTune models. Compared to RLG, FLATCHAT trades off a small portion of performance for significantly improved efficiency, with two orders of magnitude, seconds vs. hours. When batch size is increased to more the 1,500 tokens, RLG desperately fails as the number of word types grows to more than 512, which is the rank of the gradient matrix for MT models (see also in Appendix D). The inference time on a single large batch is also exacerbated to more than a day, which hinders rational attackers from considering RLG. In contrast, FLATCHAT only has a small performance drop on very large batches, and all corresponding

---

[3]Hugging Face GPT-2: `https://huggingface.co/gpt2`.

[4]Following standard nomenclature, *types* denotes the unique words, while *tokens* denote word instances.

[5]Using a learning rate $5 \times 10^{-4}$ and momentum 0.9.

[6]For a fair comparison, both attacks are tested on CPU servers with the same hardware environment.

[7]We also consider polynomial regression with a higher degree, please see Appendix B.

Table 2: The attack performance of FLATCHAT and RLG on IWSLT training batches with their overall token sizes in $\{100, 500, 1000, 1500, 2000\}$. A linear regression model is used to predict the number of word types in $\mathcal{B}$.

| Method | #Token | #Type | Scratch F-1 (ORAC.) | Prec. | Recall | F-1 | Time | FineTune #Type | F-1 (ORAC.) | Prec. | Recall | F-1 | Time |
|---|---|---|---|---|---|---|---|---|---|---|---|---|---|
| FLATCHAT | 100 | 52.3 | 0.9556 | 0.9364 | 0.9576 | 0.9469 | 0.7s | 58.5 | 0.8233 | 0.7936 | 0.8108 | 0.8021 | 0.8s |
| RLG | | | 0.9729 | 1.0000 | 0.9729 | 0.9863 | 1.0h | | 0.9980 | 0.9981 | 1.0000 | 0.9990 | 1.9h |
| FLATCHAT | 500 | 214.2 | 0.9190 | 0.8826 | 0.9188 | 0.9003 | 1.1s | 204.6 | 0.8006 | 0.7851 | 0.7896 | 0.7874 | 1.1s |
| RLG | | | 0.9421 | 1.0000 | 0.9421 | 0.9702 | 19.3h | | 0.9733 | 0.9742 | 1.0000 | 0.9869 | 23.6h |
| FLATCHAT | 1000 | 379.2 | 0.8514 | 0.8525 | 0.8523 | 0.8524 | 3.5s | 419.6 | 0.7751 | 0.8616 | 0.6041 | 0.7102 | 1.7s |
| RLG | | | 0.9956 | 1.0000 | 0.9956 | 0.9978 | 36.1h | | 0.8060 | 0.9933 | 0.8060 | 0.8899 | 33.6h |
| FLATCHAT | 1500 | 603.3 | 0.8253 | 0.882 | 0.7812 | 0.828 | 3.1s | 577.1 | 0.7683 | 0.8484 | 0.5535 | 0.6699 | 1.3s |
| RLG | | | 0.0160 | 1.0000 | 0.0160 | 0.0315 | 33.1h | | 0.0112 | 0.8750 | 0.0112 | 0.0222 | 33.6h |
| FLATCHAT | 2000 | 762.8 | 0.781 | 0.9231 | 0.671 | 0.7771 | 1.5s | 723.9 | 0.7505 | 0.8483 | 0.5331 | 0.6548 | 1.5s |
| RLG | | | 0.0051 | 1.0000 | 0.0051 | 0.0101 | 34.3h | | 0.0064 | 1.0000 | 0.0064 | 0.0128 | 37.6h |

Table 3: The attack performance on batches with the shape seen (8 x 25, 16 x 50 and 32 x 100) or unseen (64 x 50 and 128 x 100) in training. A linear regression model is used to predict the number of unique word types in batches.

| Test | IMDB | | | | | | AGNEWS | | | | | |
|---|---|---|---|---|---|---|---|---|---|---|---|---|
| Shape of $\mathcal{B}$ | #Type | MAER | Prec. | Recall | F-1 | F-1 (ORAC.) | #Type | MAER | Prec. | Recall | F-1 | F-1 (ORAC.) |
| 8 x 25 | 140.8 | 0.3026 | 0.7098 | 0.8655 | 0.7800 | 0.8257 | 149.4 | 0.3605 | 0.6731 | 0.8938 | 0.7679 | 0.8536 |
| 16 x 50 | 426.3 | 0.3281 | 0.6756 | 0.8926 | 0.7691 | 0.8263 | 447.3 | 0.3468 | 0.6689 | 0.8825 | 0.7610 | 0.8254 |
| 32 x 100 | 1169.0 | 0.1102 | 0.7625 | 0.8454 | 0.8018 | 0.8086 | 936.0 | 0.1811 | 0.7420 | 0.8370 | 0.7867 | 0.8086 |
| 64 x 50 | 1211.5 | 0.0635 | 0.7805 | 0.8270 | 0.8031 | 0.8064 | 1439.5 | 0.0786 | 0.7856 | 0.7979 | 0.7917 | 0.7968 |
| 128 x 100 | 3152.4 | 0.3267 | 0.8607 | 0.5790 | 0.6923 | 0.7875 | 2684.8 | 0.2282 | 0.8356 | 0.6410 | 0.7254 | 0.7788 |

attacks on average are finished within several seconds.[8] The attack performance on FineTune is less significant than those on Scratch. We attribute this to the smaller expected absolute values of the gradients given better model predictions.

### 4.2.2 EXPERIMENTS ON LARGE-SCALE LANGUAGE MODELS.

LLM experiments focus on testing the ability of FLATCHAT on a large-scale setting. Note that $|\mathcal{V}|$ in GPT-2 is larger than 50 thousand, where a single RLG attack is estimated to cost a couple of days. We compare FLATCHAT on a test set with batch shapes seen in training, *i.e.,* $8 \times 25$, $16 \times 50$ and $32 \times 100$. The overall attack performance is considered quite positive given its ability to detect less than 1/50 word types in a large vocabulary and achieve over 75% F-1 scores. Furthermore, after increasing the batch size to $128 \times 100$, which is about four times the scale of the largest batches considered in training the type number estimator, the attack performance is still promising, resulting in around 70% F-1. The mean absolute error ratio (MAER) to the oracle type size shows the estimated type number is rational even on batches with unseen shapes. To sum up, LLM experiments show that our attack is robust in multiple scenarios, including *i)* large vocabulary; *ii)* batches with large sizes unseen in training; and *iii)* transferred domains that differ from the attacker's training dataset.

### 4.2.3 ANALYTICAL STUDY

**Comparison of ranking scorers.** We compare FLATCHAT based on two scorers *Abs.* and *GMM* in Table 4. *GMM* consistently provides better ranking scores on the used word types than *Abs.* as *GMM* uses the estimated distributions of both positive and negative clusters effectively. The attacks using a linear estimator to predict word type numbers also achieve comparable performance to those using oracle size numbers on both *Abs.* and *GMM* scorers.

---

[8]The running time varies as sometimes the Gaussian Mixture Model is sensitive to initialization and we re-run *GMM* until the positive cluster has a significantly larger variance than the negative one.

Table 4: The comparison of FLATCHAT using *Abs.* and *GMM* as ranking scores given ground truth (ORACLE) or predicted (PREDICT) $|\mathcal{T}|$. The batch shape is set to $32 \times 100$.

| Method | IMDB | | | AGNEWS | | |
|---|---|---|---|---|---|---|
| | Prec. | Recall | F-1 | Prec. | Recall | F-1 |
| FLATCHAT (*Abs.*, ORACLE) | 0.8029 | 0.8029 | 0.8029 | 0.8039 | 0.8039 | 0.8039 |
| FLATCHAT (*GMM*, ORACLE) | **0.8086** | **0.8086** | **0.8086** | **0.8086** | **0.8086** | **0.8086** |
| FLATCHAT (*Abs.*, PREDICT) | 0.7573 | 0.8396 | 0.7963 | 0.7371 | 0.8315 | 0.7815 |
| FLATCHAT (*GMM*, PREDICT) | **0.7625** | **0.8454** | **0.8018** | **0.7420** | **0.8370** | **0.7867** |

### 4.2.4 ANALYSIS ON DEFENSE METHODS

Here, we investigate two defense mechanisms on FLATCHAT: *i)* freezing the last layer by ceasing its gradient calculation (Freeze) (Gupta et al., 2022) and *ii)* using Differentially Private Stochastic Gradient Decent (DP-SGD) (Abadi et al., 2016). Freeze is a strong defense method as all the corresponding gradients are no longer available for any attacks, however, we observe that freezing the last layer consistently introduces a cost to model performance, by comparing Last vs. Vanilla and Last vs. Last+Emb in Figure 4. DP-SGD clips the gradient vectors to a bound $C$ and adds Gaussian noise with multiplier $\sigma$. Adam (Kingma and Ba, 2014) is applied in MT experiments for fast convergence. We observe that a weak DP (with small noise $\sigma = 10^{-3}$ and $C = 1.0$) can protect the gradients from our attack in Table 5 and at the same time improve the robustness of training as illustrated in Figure 4.

Table 5: The comparison of DP-SGD with various noise multipliers $\sigma$ on training language model. We test on the batches of $b = 32 \times l = 100$ for IMDB and AGNEWS, and $b = 32$ for IWSLT. The clipping bound $C$ is set to 1.0 for all DP experiments.

| Dataset | DP-SGD ($\sigma$) | | | |
|---|---|---|---|---|
| LLM | 0.0 | $10^{-4}$ | $10^{-3}$ | $10^{-2}$ |
| IMDB | 0.7964 | 0.7969 | 0.6869 | 0.0537 |
| AGNEWS | 0.7315 | 0.7752 | 0.6935 | 0.0035 |
| MT | 0.0 | $10^{-9}$ | $10^{-8}$ | $10^{-7}$ |
| Scratch | 0.9260 | 0.7912 | 0.5304 | 0.0439 |
| MT | 0.0 | $10^{-5}$ | $10^{-4}$ | $10^{-3}$ |
| FineTune | 0.7127 | 0.6965 | 0.2947 | 0.0482 |

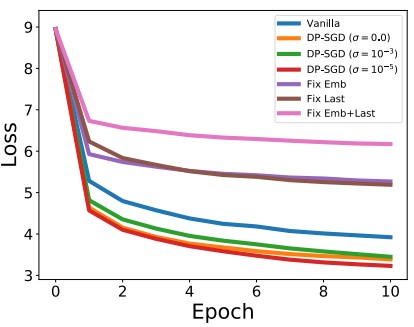

Figure 4: Comparison of validation loss of MT models with different defenses using FineTune. We plot DP-SGD with various noise multipliers $\sigma$ and Freeze the last and embedding layers, Last, Emb and Last+Emb.

## 5 CONCLUSION

We develop a novel gradient flattening method for label inference attacks, which is both effective and efficient in attacking federated large language model training. Our research demonstrates the feasibility of successfully recovering word types in challenging settings, *i.e.,* large batch size, large vocabulary and transferred domain during attacks. We further investigate defense methods against our attack and find that adding small noise in DP-SGD can effectively mitigate the impact of this new attack. These findings motivate the future implementation of FL using DP for defense even though the models are trained in large-scale industrial settings.

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
