# A  DISTRIBUTION OF VALUES $s_t$ IN FLATTENED GRADIENT VECTOR s

We illustrate the distribution of elemental values $s_t$ in $s$ by conducting LLM training on WIKITEXT using various batch sizes. The positive word types (orange bars) and negative word types (blue bars) are binned separately. In all plots, the positive cluster possesses a much flatter distribution than the negative cluster, *i.e.,* $\sigma_p \gg \sigma_n$, as discussed in Rmk. 1. The shape of the mixture distribution is affected by the total number of types in the batches. Larger batches usually mean higher count numbers for positive types while the distributions for the negative clusters are similar.

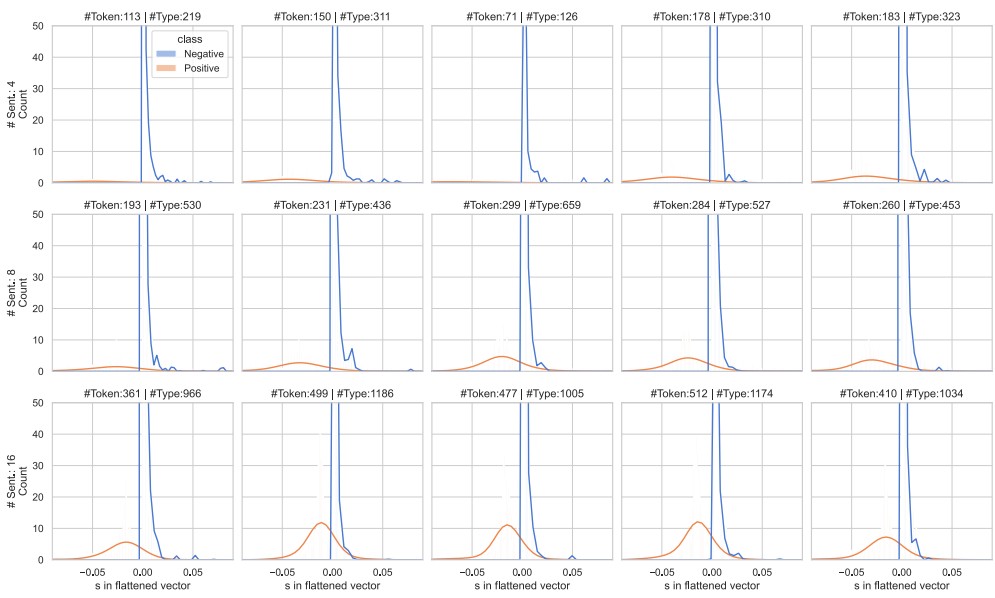

Figure 5: The comparison of the distributions of values $s$ in the flattened vectors $s$, derived from $\mathcal{B}$ with varying number of sentences in $\{4, 8, 16\}$ and max sentence length is $100$.

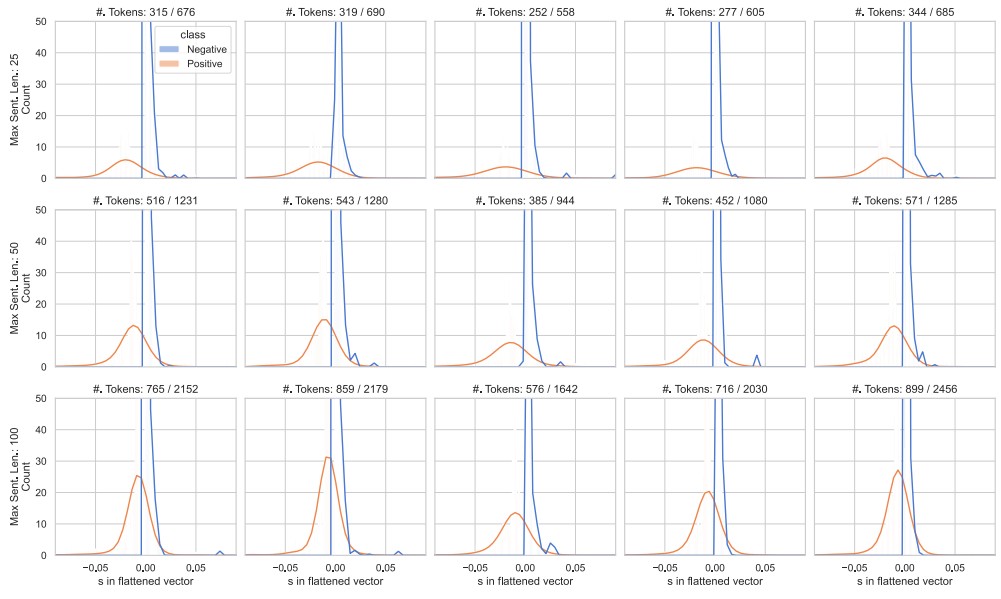

Figure 6: The comparison of the distributions of values $s$ in the flattened vectors $s$, derived from $\mathcal{B}$ with varying max sentence length in $\{25, 50, 100\}$ and number of sentence is $32$.

# B  POLYNOMIAL REGRESSION MODELS FOR PREDICTING TYPE NUMBER

We compare the performance of regression models with various degrees $d \in \{1, 2, 3, 4\}$ for predicting the number of word types $|\mathcal{T}|$ in Table 6. We consider the settings that batch shapes are seen ($8 \times 25$, $16 \times 50$ and $32 \times 100$) or unseen ($64 \times 50$ and $128 \times 100$) in LLM training. For MT experiments, we chunk the batches to within $\{100, 500, 1000, 1500, 2000\}$ total tokens. The mean absolute error ratio (MAER) is used to evaluate the difference ratio between predicted type size $\mathcal{R}(\phi_p)$ and ground-truth number of word types $|\mathcal{T}|$,

$$\text{MAER} \triangleq \frac{||\mathcal{T}| - \mathcal{R}(\phi_p)|}{|\mathcal{T}|}.$$

Table 6: The comparison of estimator based on polynomial regression model $\mathcal{R}$ with various degrees $d \in \{1, 2, 3, 4\}$. The best averaged results (Avg.) of regression models are highlighted with ‡.

| Test ($\mathcal{B}$ Shape) | #Type | $\mathcal{R}(d = 1)$ MAER | $\mathcal{R}(d = 2)$ MAER | $\mathcal{R}(d = 3)$ MAER | $\mathcal{R}(d = 4)$ MAER |
|---|---|---|---|---|---|
| 8 x 25 | 140.8 | 0.3026 | 0.1626 | 0.1625 | 0.1576 |
| 16 x 50 | 426.3 | 0.3281 | 0.4822 | 0.4835 | 0.4930 |
| 32 x 100 | 1169.0 | 0.1102 | 1.1167 | 0.9053 | 0.2200 |
| 64 x 50 | 1211.5 | 0.0635 | 1.0081 | 0.8120 | 0.1828 |
| 128 x 100 | 3152.4 | 0.3267 | 0.9391 | 0.4220 | 5.1350 |
| Avg. | 1220.0 | 0.2262‡ | 0.7417 | 0.5571 | 1.2377 |

(a) LLM- IMDB

| Test ($\mathcal{B}$ Shape) | #Type | $\mathcal{R}(d = 1)$ MAER | $\mathcal{R}(d = 2)$ MAER | $\mathcal{R}(d = 3)$ MAER | $\mathcal{R}(d = 4)$ MAER |
|---|---|---|---|---|---|
| 8 x 25 | 149.4 | 0.3605 | 0.1497 | 0.1496 | 0.1416 |
| 16 x 50 | 447.3 | 0.3468 | 0.5641 | 0.5585 | 0.5607 |
| 32 x 100 | 936.0 | 0.1811 | 0.8987 | 0.7764 | 0.3032 |
| 64 x 50 | 1439.5 | 0.0786 | 1.1548 | 0.8643 | 0.9165 |
| 128 x 100 | 2684.8 | 0.2282 | 1.1890 | 0.5965 | 5.4291 |
| Avg. | 1131.4 | 0.2390‡ | 0.7913 | 0.5891 | 1.4702 |

(b) LLM- AGNEWS

| Test (#Token) | #Type | $\mathcal{R}(d = 1)$ MAER | $\mathcal{R}(d = 2)$ MAER | $\mathcal{R}(d = 3)$ MAER | $\mathcal{R}(d = 4)$ MAER |
|---|---|---|---|---|---|
| 100 | 52.3 | 0.0487 | 0.0642 | 0.0434 | 0.0458 |
| 500 | 214.2 | 0.0756 | 0.0885 | 0.0596 | 0.0549 |
| 1000 | 379.2 | 0.0557 | 0.0623 | 0.0623 | 0.0641 |
| 1500 | 577.1 | 0.1287 | 0.1242 | 701.3872 | 0.1338 |
| 2000 | 762.8 | 0.2699 | 0.2645 | 534.2593 | 6476.1500 |
| Avg. | 402.4 | 0.1157‡ | 0.1207 | 247.1624 | 1295.2900 |

(c) MT- Scratch

| Test (#Token) | #Type | $\mathcal{R}(d = 1)$ MAER | $\mathcal{R}(d = 2)$ MAER | $\mathcal{R}(d = 3)$ MAER | $\mathcal{R}(d = 4)$ MAER |
|---|---|---|---|---|---|
| 100 | 58.8 | 0.1884 | 0.1968 | 0.1401 | 0.2531 |
| 500 | 204.6 | 0.1312 | 0.1337 | 0.1776 | 0.2090 |
| 1000 | 419.6 | 0.2466 | 0.2397 | 0.2874 | 0.2752 |
| 1500 | 577.1 | 0.3367 | 0.3324 | 0.3433 | 0.3310 |
| 2000 | 723.9 | 0.3728 | 0.3693 | 0.3652 | 0.3568 |
| Avg. | 396.7 | 0.2552 | 0.2544‡ | 0.2627 | 0.2850 |

(d) MT- FineTune

Table 7: Comparison of attack performance using polynomial regression models $\mathcal{R}$ with various degrees $d \in \{1, 2, 3\}$.

| Test | IMDB | | | AGNEWS | | |
|---|---|---|---|---|---|---|
| ($\mathcal{B}$ Shape) | Prec. | Recall | F-1 | Prec. | Recall | F-1 |
| Using oracle number of word types $\|\mathcal{T}\|$ | | | | | | |
| 8 x 25 | 0.8257 | 0.8257 | 0.8257 | 0.8536 | 0.8536 | 0.8536 |
| 16 x 50 | 0.8263 | 0.8263 | 0.8263 | 0.8254 | 0.8254 | 0.8254 |
| 32 x 100 | 0.8086 | 0.8086 | 0.8086 | 0.8086 | 0.8086 | 0.8086 |
| 64 x 50 | 0.8064 | 0.8064 | 0.8064 | 0.7968 | 0.7968 | 0.7968 |
| 128 x 100 | 0.7875 | 0.7875 | 0.7875 | 0.7788 | 0.7788 | 0.7788 |
| Using predicted $\|\mathcal{T}\|$ by $\mathcal{R}(d=1)$ | | | | | | |
| 8 x 25 | 0.7098 | 0.8655 | 0.7800 | 0.6731 | 0.8938 | 0.7679 |
| 16 x 50 | 0.6756 | 0.8926 | 0.7691 | 0.6689 | 0.8825 | 0.7610 |
| 32 x 100 | 0.7625 | 0.8454 | 0.8018 | 0.7420 | 0.8370 | 0.7867 |
| 64 x 50 | 0.7805 | 0.8270 | 0.8031 | 0.7856 | 0.7979 | 0.7917 |
| 128 x 100 | 0.8607 | 0.5790 | 0.6923 | 0.8356 | 0.6410 | 0.7254 |
| Using predicted $\|\mathcal{T}\|$ by $\mathcal{R}(d=2)$ | | | | | | |
| 8 x 25 | 0.8065 | 0.7995 | 0.8030 | 0.8030 | 0.8643 | 0.8325 |
| 16 x 50 | 0.6171 | 0.9037 | 0.7334 | 0.5994 | 0.8970 | 0.7186 |
| 32 x 100 | 0.4514 | 0.9497 | 0.6120 | 0.5003 | 0.9101 | 0.6456 |
| 64 x 50 | 0.4671 | 0.9331 | 0.6226 | 0.4378 | 0.9200 | 0.5932 |
| 128 x 100 | 0.5098 | 0.9222 | 0.6566 | 0.4612 | 0.9124 | 0.6127 |
| Using predicted $\|\mathcal{T}\|$ by $\mathcal{R}(d=3)$ | | | | | | |
| 8 x 25 | 0.8101 | 0.7959 | 0.8030 | 0.8052 | 0.8608 | 0.8321 |
| 16 x 50 | 0.6166 | 0.9042 | 0.7332 | 0.5997 | 0.8966 | 0.7187 |
| 32 x 100 | 0.4965 | 0.9422 | 0.6503 | 0.5272 | 0.9043 | 0.6661 |
| 64 x 50 | 0.5126 | 0.9256 | 0.6598 | 0.4952 | 0.9084 | 0.6410 |
| 128 x 100 | 0.6415 | 0.8755 | 0.7404 | 0.5757 | 0.8757 | 0.6947 |

(a) Large Language Model (LLM)

| Test | Scratch | | | FineTune | | |
|---|---|---|---|---|---|---|
| (#Tokens) | Prec. | Recall | F-1 | Prec. | Recall | F-1 |
| Using oracle number of word types $\|\mathcal{T}\|$ | | | | | | |
| 100 | 0.9556 | 0.9556 | 0.9556 | 0.8233 | 0.8233 | 0.8233 |
| 500 | 0.9190 | 0.9190 | 0.9190 | 0.8006 | 0.8006 | 0.8006 |
| 1000 | 0.8514 | 0.8514 | 0.8514 | 0.7751 | 0.7751 | 0.7751 |
| 1500 | 0.7912 | 0.7912 | 0.7912 | 0.7683 | 0.7683 | 0.7683 |
| 2000 | 0.7810 | 0.7810 | 0.7810 | 0.7505 | 0.7505 | 0.7505 |
| Using predicted $\|\mathcal{T}\|$ by $\mathcal{R}(d=1)$ | | | | | | |
| 100 | 0.9364 | 0.9576 | 0.9469 | 0.7936 | 0.8108 | 0.8021 |
| 500 | 0.8826 | 0.9188 | 0.9003 | 0.7851 | 0.7896 | 0.7874 |
| 1000 | 0.8525 | 0.8523 | 0.8524 | 0.8616 | 0.6041 | 0.7102 |
| 1500 | 0.8820 | 0.7812 | 0.8280 | 0.8484 | 0.5535 | 0.6699 |
| 2000 | 0.9231 | 0.6710 | 0.7771 | 0.8483 | 0.5331 | 0.6548 |
| Using predicted $\|\mathcal{T}\|$ by $\mathcal{R}(d=2)$ | | | | | | |
| 100 | 0.7947 | 0.7994 | 0.7971 | 0.9902 | 0.9395 | 0.9642 |
| 500 | 0.7810 | 0.7933 | 0.7871 | 0.8698 | 0.9201 | 0.8942 |
| 1000 | 0.8494 | 0.6450 | 0.7332 | 0.8426 | 0.8559 | 0.8492 |
| 1500 | 0.8501 | 0.5685 | 0.6814 | 0.8778 | 0.7839 | 0.8282 |
| 2000 | 0.8486 | 0.5364 | 0.6573 | 0.9222 | 0.6751 | 0.7796 |
| Using predicted $\|\mathcal{T}\|$ by $\mathcal{R}(d=3)$ | | | | | | |
| 100 | 0.9642 | 0.9556 | 0.9599 | 0.8384 | 0.7573 | 0.7958 |
| 500 | 0.9448 | 0.9052 | 0.9246 | 0.8349 | 0.7298 | 0.7788 |
| 1000 | 0.8644 | 0.8458 | 0.8550 | 0.8602 | 0.6117 | 0.7150 |
| 1500 | 0.7725 | 0.7289 | 0.7501 | 0.8477 | 0.5583 | 0.6732 |
| 2000 | 0.8284 | 0.6045 | 0.6989 | 0.8439 | 0.5369 | 0.6562 |

(b) Machine Translation (MT)

Then, we test FLATCHAT using different regression models in Table 7. The attack performance is strongly correlated to the type number estimator. Linear regression model, $\mathcal{R}(d = 1)$, on average achieves the most stable and the best averaged performance compared with high-order polynomial regression models ($d \geq 2$). Notably, $\mathcal{R}(d = 3)$ and $\mathcal{R}(d = 4)$ sometimes diverge on MT-Scratch as shown in Table 6.c. The linear models are used as type number estimators in our main paper.

## C   PROOF OF SPARSE VALUES IN FLATTENED GRADIENT VECTOR s

In this section, we ground the rationality of using the absolute value of $s_t$ (*Abs.*) to infer word type usage. *Abs.* is considered as a baseline to *GMM* (Rmk.3), compared in Section 4.2.3.

**Theorem 5 (Sparsity of $\boldsymbol{g}$.)**  *For the gradient vector $\boldsymbol{g}_i$ regarding the output token / label $y_i$, only one element's absolute value $|g_{y_i,i}|$ is significantly larger than the absolute values of any other elements $t$ in $\boldsymbol{g}_i$,*

$$|g_{y_i,i}| \gg |g_{t,i}|, \;\; \text{if } t \neq y_i. \tag{14}$$

**Proof**  *We first calculate the gradient vector $\boldsymbol{g}_i$ on the $i$-th incidence, where $\boldsymbol{p}_i$ is the output probability vector by the language model and $\boldsymbol{y}_i$ is a one-hot vector with the $t$-th element to be one and other elements equal to zero,*

$$\boldsymbol{g}_i \triangleq \frac{\partial \mathcal{L}}{\partial \boldsymbol{z}_i} = \boldsymbol{p}_i - \boldsymbol{y}_i. \tag{15}$$

*Taking the sum of the gradient values, we show the absolute gradient value regarding the ground-truth token is the sum of the absolute values by others,* i.e., $|g_{y_i,i}| = \sum_{t \neq y_i} |g_{t,i}|$,

$$g_{y_i,i} + \sum_{t \neq y_i} g_{t,i} = (p_{y_i,i} - 1) + \sum_{t \neq y_i} p_{t,i} = \sum_{t} p_{t,i} - 1 = 1 - 1 = 0. \tag{16}$$

*Then, we have*

$$|g_{y_i,i}| = \Big| \sum_{t \neq y_i} g_{t,i} \Big|. \tag{17}$$

*Because, $\forall\, t \neq y_i, g_{t,i} = p_{t,i} \geq 0$, as probabilistic outputs are larger or equal to zero,*

$$|g_{y_i,i}| = \sum_{t \neq y_i} |g_{t,i}|. \tag{18}$$

*Considering the huge number of unused tokens $|\mathcal{V}| - 1$, we have Eqn. 14.*

**Remark 4 (*Abs.* scorer on $s_t$)**  *We can approximately consider gradient vectors $\boldsymbol{g}_i$ as scaled 'one-hot' vectors with the $t$-th element significantly larger than other elements, where $t = y_i$. Then, the flattened vector $\boldsymbol{s}$ is a weighted combination of these sparse gradient vectors with the used types usually possess much larger absolute values, $|s_t|$ where $t \in \mathcal{T}$, than the absolute values regarding the unused types, $|s_n|$ where $n \notin \mathcal{T}$. Ranking $s_t$ provides clues to the used word types.*

## D   TRAINING LANGUAGE MODELS

**The architectures of victim language models.**    The architectures of the neural network used as victim models for MT and LLM experiments are summarized in Table 8 and 9. More details about the implementation of the model basis are at FAIRSEQ and HuggingFace.

**Computational environment for Attackers.**    To conduct fair comparison, FLATCHAT and RLG use the same experimental environment, CPU servers with Intel(R) Xeon(R) Gold 6154 CPU @ 3.00GHz. All computations are based on basic NumPy package in Python without specific optimization to the algorithms. Note that *i)* FLATCHAT can be easily sped up by using GPU to compute the matrix operations; and *ii)* Running time of RLG grows in a linear manner in terms of search rounds, but there is no notable improvement in attack performance.

Table 8: The setting of victim models used in MT experiments.

| Model Type | Transformer |
|---|---|
| Embedding Dimension | 512 |
| Number of Heads | 4 |
| Number of Layer | 6 |
| Attention Dropout Rate | 0.3 |
| Embedding Dropout Rate | 0.3 |
| Vocabulary Size | 16,594 |

Table 9: The setting of victim models used in LLM experiments.

| Model Type | GPT-2 |
|---|---|
| Embedding Dimension | 768 |
| Number of Heads | 12 |
| Number of Layer | 12 |
| Attention Dropout Rate | 0.1 |
| Embedding Dropout Rate | 0.1 |
| Vocabulary Size | 50,257 |

## E  PERFORMANCE OF MACHINE TRANSLATION MODELS WITH DEFENSE

The performance of machine translation models is evaluated based on IWSLT 2017 test set using scareBLEU. We validate the models with various defense settings such as freezing different layers (Last, Emb and Last+Emb) and DP-SGD with varying noise multipliers $\sigma \in \{0.0, 10^{-3}, 10^{-5}\}$, as addressed in Table 10. The main results are aligned with those of losses on the validation set reported in Figure 4. Although freezing layers provide strong defense against gradient inference attacks on corresponding layers, the performance of the models drops significantly. In contrast, DP-SGD with small noise achieves highly competitive results with the vanilla model and clipping gradients even encourages more robust training.

Table 10: The comparison of machine translation models trained with different defense settings. For each setting, two checkpoints at epoch 10 and epoch 50 are tested.

| Model | Vanilla | Freeze | | | DP-SGD | | |
|---|---|---|---|---|---|---|---|
| | | Emb | Last | Emb+Last | $\sigma = 0.0$ | $\sigma = 10^{-3}$ | $\sigma = 10^{-5}$ |
| Epoch 10 | 14.2 | 1.9 | 8.2 | 1.2 | 16.6 | 15.0 | 17.8 |
| Epoch 50 | 16.7 | 2.4 | 9.9 | 3.6 | 17.6 | 18.9 | 20.6 |