# OpenReview forum: "FLAT-Chat: A Word Recovery Attack on Federated Language Model Training"
_ICLR.cc/2024/Conference — Submitted to ICLR 2024_

### Official Review · Reviewer_3Q3Y · 2023-10-30

**Soundness:** 2 fair
**Presentation:** 1 poor
**Contribution:** 2 fair
**Rating:** 3
**Confidence:** 5

**Summary:**

The paper proposes a new gradient label leakage procedure. The procedure "flattens" the gradients of the last linear layer of the network and decomposes it into two terms corresponding to samples that are correctly classified and those that are not. Each term is approximated with a Gaussian whose unknown parameters are fitted jointly with a GMM on additional data. Then, each possible label is ranked based on its likelihood of being present in the data batch calculated using the parameters of those Gaussians. Finally, the total number of different labels present in the batch is estimated based on linear regression over the weights of the two Gaussian weight factors. This in combination with the ranking, produces the set of labels present in the data batch. The authors apply this technique to federated learning of LLMs and machine translation algorithms to leak the set of tokens that are used to train the Transformer models. The authors demonstrate this procedure results in 0.7/0.8 F1 score for large batches of many individual tokens with realistic-sized vocabularies.

**Strengths:**

- Experiments on fairly large models ( GPT-2 )
- Experiments on large sequences and batches
- The use of GMM is interesting

**Weaknesses:**

- **The description of the proposed method can be hard to read at times:**
I know a lot about this particular area of research and I still struggled to follow the presentation of Section 3 (the technical contribution section). To this end, in my opinion, the paper will really benefit from a paragraph (probably coupled with a summary figure) that summarizes the steps of the proposed method early on in Section 3, so that it is easier to follow what the paper is trying to achieve through the different subsections of Section 3. It needs not be long, consider something like the beginning of my paper summary above. Similarly, presenting the full algorithm at the end of Section 3 will help a lot in understanding how the different pieces of the algorithm fit together. Further, the paper will also benefit from giving more intuitive explanations of its steps throughout. One example of this will be to present Eq. 9 before Theorem 3 to make it intuitively clear where the GMM pieces come from in Eq.6. Finally, there are several key missing from Section 3. It should explicitly state that the GMMs and the regression model on $\|\mathcal{T}\|$ need to be fitted on auxiliary data, and how estimating the number of unique tokens from the regression model is used together with the ranking to provide the set of recovered tokens. It should also state that during the LLM training, when multiple tokens are predicted, the CE loss is summed across all of them which mathematically is equivalent to the label recovery from a large batch.
- **Citating and comparing to prior work:**
The paper should cite and compare against prior label reconstruction attacks outside of RLG [5]. In particular, [1] can be used for recovering the set of unique tokens, while [2,3] can be used to recover the counts as well. Comparing against [1-3] is absolutely crucial, in my opinion, for accepting this paper, as those methods would work fast for large vocabularies and long sentences, unlike RLG, and have been shown to be effective at recovering labels to very good accuracy. Further, [2], in particular, is very closely related to FLAT-CHAT, as it derives the same "flattening" operation the authors claim as a contribution in the text. To this end, the authors should not claim the flattening operation as a contribution and instead clearly mark the derivation presented there as equivalent to the one made in [2].
Given the similarities to prior work, the authors should also consider including an explicit discussion of how their method differs from prior work. Finally, the authors acknowledge that FILM [4] can be applied to the same problem the authors consider but from the input side of the network. Yet, they do not provide a comparison. While beating it is not required for acceptance (due to the different requirements the attacker has), comparing against it is a good idea.
- **The attack setting:**
Label leakage attacks like [2] and [3], are capable not only of recovering the set of unique tokens in input data but also their counts. The authors should provide a discussion on whether counts are important from LLM privacy point of view.
Further, the authors should better motivate their attacker's goal in general. While privacy is indeed violated by knowing the set of tokens fed to the network from a purely theoretical point of view, I would reasonably think that a large percent of the vocabulary tokens occur in a large batch of long excerpts of text anyway, and when the recovery has a precision of 0.85 and recall of 0.5 it will be very hard from a practical perspective to gain any reasonable sensitive information. That is, I expect the rank of rare tokens, which tend to be more private, to be lower in your method due to their lower occurrence rates. I also expect that the recall will be much lower than 0.5 for labels that are in the middle of the ranking.  Thus, in such a situation, the attacker will obtain that words like "the", "I", "you" are present in the batch with high accuracy, but will rarely obtain, let's say, a phone number. The problem gets even worse when considering the fact that LLMs are trained on tokens and not full words.
- **Bad evaluation results:**
The results shown in the experiment do not convince me in the superiority of the proposed method. In particular, RLG consistently and by big margins results in better reconstructions than FLAT-CHAT if RLG is in the mode where it is applicable (\|\mathcal{T}\| < D). [1-3], which do not have such restrictions and tend to work much faster than RLG, might, therefore, turn out to be much better than FLATCHAT.
 Even outside of these concerns, I find the precision of 0.85 and the recall of 0.5 in Table 2 and the 0.7 precision and 0.85 recall numbers in Table 3 not that convincing in terms of their practical attack relevance as outlined above.
- **Suggestions:**
1. Essentially, the method proposes to model the $p_{i,j}$ as Gaussian distribution, which as $\|\mathcal{B}\|->\infty$ get closer to the truth but since $0\leq p_{i,j}\leq 1$ is a probability the approximation for finite $\|\mathcal{B}\|$ is very bad. This is also reflected in the authors' shown negative clusters in the figures of Appendix A. The authors can consider modeling the $\log p_{i,j} $ as Gaussian ( but $\alpha_i$ still as Gaussian ). In my quick tests, this reflected the shown negative cluster pdf shapes much better.
2. The authors propose to use Equation 12 as a ranking function. If a proper prior $p(s_t|t \in \mathcal{B})$ is used, Equation 12 can be used as a decision criterion instead, which will eliminate the need for using regression to fit $\|\mathcal{T}\|$. This can possibly improve the performance of the method.
- **Nits:**
1. In the first part of Eq. 13, $\sigma_n$ and $\sigma_p$ should be switched in the normalization constants of the Gaussians
2. Equations 6 and 9 assume sum instead of mean gradient aggregation. Equation 7 assumes a  mean instead of sum. This needs to be made consistent throughout the paper.

**Questions:**

- [Crucial] Can the authors provide a comparison to [1-3]? Can the authors provide an explanation of why they are better than [1-3] if they are?
- Can you provide a comparison to FILM [4]?
- Can you explain what auxiliary data was used to obtain the parameters of FLAT-CHAT in Table 2 (Machine Translation) experiments?
- Can you explain why the precision and recall numbers between Tables 2 and 3 differ that much?
- Can the authors explain why approximating $\|\mathcal{T}\|$ separately is needed? Wouldn't using the optimal Bayesian criterion with a prior ratio of $\frac{\|\mathcal{B}\|}{(\|\mathcal{V}\|-1)\|\mathcal{B}\|}$ be sufficient?
- Can the approximation of $\|\mathcal{T}\|$ be improved by using some of the methods in [1-3] - it seems that currently, the approximation is far from perfect, to the point it has a few % difference on the final performance?
- Can the authors explain the reasoning behind the Abs baseline in Appendix C? Seems that what the authors propose there is very similar to [1] - what are the similarities and differences?
- Can you provide precise runtimes of the proposed method and baselines?
- [Not so important] Can the authors run their experiments on a newer open-source LLM like Llama [6] or Chinchilla [7]?
- [Not so important] Can the authors adapt their method to model the probabilities $p_{i,j}$ with a Log-Gaussian distribution?

All in all, the paper suffers from too many issues to be accepted right now. First and most importantly, it fails to compare to relevant prior work that has a reasonable chance to work better in practice than the proposed method and claims as a contribution the derivation of the "flattening" operation on the gradient despite the fact it is known. Second, the paper is hard to follow due to a lack of method summary and intuitive explanations. Finally, the paper needs to spend more time justifying the problem setting and their results in the context of this setting, as currently, I am not sure if the privacy concerns raised by the proposed attack are realistic.

[1] Yin, Hongxu, et al. "See through gradients: Image batch recovery via gradinversion." Proceedings of the IEEE/CVF Conference on Computer Vision and Pattern Recognition. 2021.
[2] Wainakh, Aidmar, et al. "User-level label leakage from gradients in federated learning." arXiv preprint arXiv:2105.09369 (2021).
[3] Geng, Jiahui, et al. "Towards general deep leakage in federated learning." arXiv preprint arXiv:2110.09074 (2021).
[4] Samyak Gupta, Yangsibo Huang, Zexuan Zhong, Tianyu Gao, Kai Li, and Danqi Chen. 2022. Recovering private text in federated learning of language models. In Advances in Neural Information Processing Systems
[5] Trung Dang, Om Thakkar, Swaroop Ramaswamy, Rajiv Mathews, Peter Chin, and Françoise Beaufays, 2021. Revealing and protecting labels in distributed training. Advances in Neural Information Processing Systems, 34:1727–1738.
[6] Touvron, Hugo, et al. "Llama: Open and efficient foundation language models." arXiv preprint arXiv:2302.13971 (2023).
[7] Hoffmann, Jordan, et al. "Training compute-optimal large language models." arXiv preprint arXiv:2203.15556 (2022).

**Details Of Ethics Concerns:**

It is not strictly needed but the paper will benefit from an Ethics statement where the authors can explain what the implications of the proposed attack are to real FL setups and emphasize the proposed solution of using differential privacy.

---

> ### Author Response · Authors · 2023-11-23
> **Response to Reviewer 3Q3Y**
>
> Thank you for your comments and suggestions. We hope the following clarifications can address your concerns.
>
> ---
>
> **W1:** The description of the proposed method can be hard to read at times, suggesting reorganizing paragraphs and the algorithm in section 3 and including some missing information (e.g. GMM fitting, regression prediction and training loss constitution).
>
> **A1:** We sincerely appreciate these suggestions regarding rearrangements and supplements. We will incorporate them in our revision.
>
> ---
>
> **W2:** Citation and comparison to related work: [1-4].
>
> **A2:** Thank you for pointing out the literature.
>
> [1] and [3] are for image reconstruction, focusing on image data in a continuous space while our work focuses on text data in a discrete space.
>
> [2] shares similar insights to our Lemmas 1 and 2. We have compared this baseline (coined Abs.) in Table 4. We will acknowledge our Abs. is equivalent to the suggested work. We would like to highlight our progress compared with [2]: (1) we have extended the theory and method (see GMM score and its results) and (2) our approach manages to estimate the number of tokens used in the recent training batch, which is often omitted in many related works.
>
> [4]  FILM targets sequence recovery while our work focuses on *industrial-scale* word recovery attack. Nonetheless, our work could serve as the first *Bag-of-Words* Extraction step of FILM.
>
> ---
>
> **W3:** Considering not only recovering the unique tokens but also counts (frequency) in the attack setting from a practical perspective.
>
> **A3:** We argue recovering the appearance of tokens is the first and the most essential step for information leakage. The frequency of a token is arguably less important than know it is used or not. Nonetheless, we appreciate the suggestion and would like to leave it to future work.
>
> ---
>
> **W4:** Bad evaluation results, compared with RLG in its applicable scenarios, or methods [1-3] with less restrictions. Besides, some presented precision-recall results.
>
> **A4:** Actually, the out-of-scope size of training data is quite common, imagining industrial FL scenarios using large training batches. DLG is not aware of the potential issue of a numerical value exceeding the range of $D$ which is usually several hundred. Our estimation model based on the parameters from GMM could solve the issue very well, according to Figure 3.
>
> ---
>
> **S5:** Adapt modeling the probabilities pi,j with a Log-Gaussian distribution (log pi,j).
>
> **A5:** Thanks for the suggestion, this method may serve as a better estimator. We will consider adding this to our future work.
>
> ---
>
> **S6:** Can the approximation of $|T|$ be improved by using $\frac{|B|}{(|V|-1)|B|}$ or some of the methods in [1-3]?
>
> **A6:** There is an issue in the suggested method to estimate $|T|$: the $|B|$ is the suggested formula, $\frac{|B|}{(|V|-1)|B|}$, will be canceled. We did not find promising methods in predicting $|T|$ in [1-3], especially under the setting of large batch size and large vocabulary size.
>
> ---
>
> **N7:** Eq.13, Eq.6, Eq.9 and Eq. 7 modification suggestions.
>
> **A7:** We sincerely appreciate your suggestions, and we will incorporate them into our revision.
>
> ---
>
> **Q8:** What auxiliary data was used to obtain the parameters of FLAT-CHAT in Table 2 (Machine Translation) experiments?
>
> **A8:** The auxiliary data used in Table 2 MT experiments for fine-tuning is the News-Commentary v15 de-en dataset. We will include an explanation in our revision.
>
> ---
>
> **Q9:** Can you explain why the precision and recall numbers between Tables 2 and 3 differ that much?
>
> **A9:** The results in Table 2 and Table 3 are consistent. We assume you are asking the reason why RLG may have significantly different precision and recall when *\# Tokens* is large. It is because RLG (1) fails the predict the correct size of the batch and (2) fails to predict the `*long-tail*’ tokens.
>
> ---
>
> **Q10:** Can the authors explain the reasoning behind the Abs baseline in Appendix C? the authors propose there is very similar to [1]
>
> **A10:**  Actually this methodology is similar to the one in [1], we will include the comparison and citation to our revision.
>
> ---
>
> **Q11:** Can you provide precise runtimes of the proposed method and baselines?
>
> **A11:**  The runtime depends on the batch size of the training data, and we have provided the averaged running time for comparing the efficiency of RLG and our method in Table 3. Considering some cases of 1.5 seconds vs 34.3 hours, the significance of our approach is clear.
>
> ---
>
> **Q12:** Run experiments on a newer open-source LLM like Llama [6] or Chinchilla [7].
>
> **A12:** Thanks for your suggestion. We assume our method is tested under general settings for LM which should also work for other transformer-based models including Llama and Chinchilla.

---

> ### Comment · Reviewer_3Q3Y · 2023-11-23
> **Response**
>
> The reviewer couldn't go through the comments thoroughly in the short time frame given. However, the admitted similarity of the method and results to prior work that is not even necessarily SOTA makes me reaffirm my initial grade. Further, the label recovery strategies of [1] and [3] are indeed relevant and can be applied with no modifications to the text setting. I will further read the rebuttal after the end of the discussion period.

---

### Official Review · Reviewer_CmSz · 2023-10-31

**Soundness:** 3 good
**Presentation:** 3 good
**Contribution:** 2 fair
**Rating:** 3
**Confidence:** 4

**Summary:**

This paper presents a novel attack reconstructing client's tokens in federated model training.
The authors apply two-cluster Gaussian Mixture Model(GMM) to better classify the positive tokens (those involved in training) and negative tokens, and provide a theoretical analysis proving their attack effectiveness.
Experiments on Language Modeling (LM) and Machine Translation (MT) show that FlatChat is more efficient and effective than previous method RLG.
Finally, the authors apply two defenses, FREEZE and DP-SGD, to mitigate the attack, where the former one can hurt model utility and the latter one is found an ideal solution for both model privacy and utility.

**Strengths:**

1. **Interesting research problem**. Recovering exact user input from the uploaded gradient is challenging and even harder for language model because of discrete nature of texts, so the paper has good originality.

2. **Attack with theoretical analysis**. This paper provides a new perspective from token distribution to infer the user's training texts in federated learning. The use of GMM permits infer trained tokens from gradients of a large batch of texts.

**Weaknesses:**

1. **Attack significance is low** because the order of tokens cannot be recovered. As the attack relies on gradient distribution of positive and negative tokens, the tokens' order information is hidden and not recovered. Although word distribution can leak partial privacy, in my opinion, this information is important to infer privacy underlying the training text. As a simple example, the two texts X = "A is good, B is bad", and Y = "A is bad, B is good" have the same word distribution but totally different meaning. I suggest the author to focus on or highlight specific scenarios where the word distribution can leak sufficient privacy. For example, it is possible to conduct an end-to-end case study showing how recovered tokens can lead to a more severe consequence.

2. **The technical challenge is not clear.** As Fig.2 shows, most negative tokens have vector $s$ value between 0 and 0.02. While the I appreciate the authors' efforts in visualization, it makes me doubt whether the GMM is necesssary. As the next word prediction resembles to classification, a naive baseline can apply iDLG-similar approach to directly identify (for example, with threshold) the trained words (positive tokens). Note that iDLG also leverages the last layer's gradient to infer the labels of trained samples. In this sense, the GMM is only used to better classify positives and negatives. I suggest the authors to make the attack motivation and challenges more clear in the paper.

3. **Problem importance is unclear.** From the main text, I cannot see that FL is a common solution for training/finetuning LMs, especially the large ones.  Although the authors have provided a long list of related works of training data inference attack in FL, I think it is still important to show that FL is or will be applied by organizations through real-world examples or case studies.
The only application I can imagine is using FL on mobile keyboard to predict the user's input behavior more accurately, but I'm not sure whether it trains such LMs. According to my experience, finetuning current LMs requires relative large memory, which is impractical to proceed on edge devices.
Please illustrate potential FL applications for LM training.

4. **Comparison with more baselines is needed.** I note that in Table 1 a recent work FILM also infers trained words but is not compared in Section 4.2.1. I also notice that there is a slight difference between FILM and this work in terms of $\Delta W$ but I think under FL setting the FILM can also work. Please consider compare with this attack or clarify why it is not suitable for comparison.

5. **Defense (DP-SGD) can mitigate the attack, further reducing the attack significance.** To be honest, I'm quite surprised that small noises added by DP-SGD can mitigate the attack, which is different from the conclusion in (Gupta et al. 2022). This means that this previous attack is more powerful than proposed attack because DP-SGD can not defend it without degrading the model utility.

**Questions:**

Please see my concerns in weaknesses. Besides, I also have the following questions:

1. What is the learning rate used in attack and DP-SGD? What is the resultant budget ($\epsilon$, $\delta$)?

2. What does the 'Loss' in Figure 4 mean? Training loss or validation loss?

---

> ### Author Response · Authors · 2023-11-23
> **Response to Reviewer CmSz**
>
> Thank you for your comments and suggestions. We hope the following clarifications can address your concerns.
>
> ---
>
> **W1:** Attack significance is low because the order of tokens cannot be recovered.
>
> **A1:** Revealing the training word types could also infringe serious risk for the scenarios requiring brief context, like healthcare diagnosis or personal identifiable information (PII), including credit card numbers and social security numbers. The significance of our work lies in that we are the first work that scaled up the word reconstruction attack to *industrial* settings, such as more than 10 K tokens in a batch, vocabulary size larger than 50 K and extremely fast inference speed. Furthermore, consolidated word reconstruction could serve as the first step of sequence reconstruction attacks, followed by iteratively swapping tokens and training them within a language model, as adapted in the prior work FILM and LAMP [1].
>
>
> ---
>
>
> **W2:** The technical challenge is not clear, as prior work iDLG also leverages the last layer's gradient to infer the labels of trained samples in classification tasks.
>
> **A2:**  The focus of the iDLG is to infer the input data X, and it requires gradients from all layers. The usage of last layer gradients is just to infer the classification label for better gradient matching. However, our method focuses on inferring the output data Y from a single last layer's gradients.
>
> ---
>
> **W3:** Please illustrate potential FL applications for LM training.
>
> **A3:**  The LM training can be under the scenarios like training LM for medical reports, financial reports, keyboard as mentioned etc. In many scenarios, the text corpus would be kept in clients for privacy concerns and FL will be their first choice in training shared powerful LMs (which could be used for many purposes, such as classification, QA, NLI, NLG, etc.).
>
> ---
>
>
> **W4:** Including FILM as baseline for performance comparison.
>
> **A4:**  FILM targets sequence recovery while our work focuses on *industrial-scale* word recovery attack. Nonetheless, our work could serve as the first *Bag-of-Words* Extraction step of FILM.
>
> ---
>
>
> **W5:** The attack significance is downgraded as it can be defended by DP-SGD.
>
> **A5:** The significance of our work is (1) We propose a new word recovery attack which works on *industrial-scale*, with theoretical proof and extensive empirical evidence; (2) We demonstrate the risk could be mitigated by DP-SGD. Our work enhances the motivation of using a proper security schema in FL. It is unfair to discredit a novel attack merely because it could be potentially defended, e.g., the many attack methods, FILM and LAMP [1], proposed by recent literature could also be defended by DP-SGD to some extent.
>
> ---
>
>
> **Q6:** Used learning rate and the privacy budget in DP-SGD.
>
> **A6:** Our experiments use a learning rate of $5 \times 10^{-4}$, as detailed in the footnote of Section 4.1, which introduces the experimental setup.  We will include the privacy budget in our revision.
>
> ---
>
> **Q7:** What does the 'Loss' in Figure 4 mean?
>
> **A7:** The ‘Loss’ in Figure 4 denotes the averaged cross-entropy loss, indicating the utility (how efficiently it operates in a language prediction task) of a language model. We will include an explanation in our revision.
>
> ---
>
>
> [1] LAMP: Extracting Text from Gradients with Language Model Priors (NeurIPS 2022)

---

### Official Review · Reviewer_t7LN · 2023-10-31

**Soundness:** 2 fair
**Presentation:** 2 fair
**Contribution:** 3 good
**Rating:** 5
**Confidence:** 4

**Summary:**

This paper proposes a privacy attack FLAT-Chat which recovers the set of words used in training a language model in the federated learning setting. The attack only assumes observing the gradients of the last linear layer instead of the embedding layer (as in previous work).  FLAT-Chat is inspired by the observation that the output layer gradients follow two distinct distributions for tokens used in v.s. not in training. Based on this, FLAT-Chat fits these two distributions with a two-mode Gaussian mixture, and then finds the cluster positive cluster where top K tokens are selected as the predicted training tokens. The attack is evaluated on machine translation and language modeling tasks on benchmark datasets and achieves much better attack efficiency than the previous attack Revealing Labels from Gradients (RLG).

**Strengths:**

- This attack method is novel and based on an interesting empirical observation that the gradient norm distribution is a mixture model and these two mixtures correspond to tokens in/out of the training batch.
- The attack is highly efficient and accurate as shown in Table 2, where an adversary can easily mount this attack to learn the tokens from users, demonstrating a realistic privacy concern.

**Weaknesses:**

- Some writings can be simplified, e.g. the lemmas and their proofs in Section 3.2 are simple rearrangement using some basic linear algebra which can be condensed in Equations and will not impact their readability. The theorems and the body texts are interleaved which makes the explanation of the attack less easy to follow.
- In common practice, when training language models, the parameters of the embedding layer and the last layer are typically shared, i.e. they have the same gradients. It would be a stronger attack if this more common scenario is considered.
- The attack is limited to inferring the bag of words while the order of the words cannot be recovered.

**Questions:**

- How would tying the weights between input embedding and output layer change the performance of the attack?
- Another potential defense is secure aggregation, where the server can only observe the aggregated gradients instead of individual’s. How might this impact the attack? Could the adversary still infer useful information when the set of participants is large?

---

> ### Author Response · Authors · 2023-11-23
> **Response to Reviewer t7LN**
>
> Thank you for your comments and suggestions. We hope the following clarifications can address your concerns.
>
> ---
>
> **W1:** Some writings can be simplified, e.g. the lemmas and their proofs in Section 3.2 can be condensed in Equations and will not impact their readability.
>
> **A1:** The proof of the properties is one of the key findings in this work as they ground the success of our word recovery attack. We sincerely appreciate your suggestions, and we will rearrange this content in our revision for better readability.
>
> ---
>
> **W2:** It would be better to consider the common practice of the embedding layer and the last layer sharing the same parameters in a language model.
>
> **A2:**  Yes, we have conducted tests on this setting in our preliminary study on MT tasks and our method has demonstrated successful outcomes. However, it is important to note that the share-embedding setting exposes both embedding and the last linear layers to data reconstruction attacks.
>
> ---
>
> **W3:**  The attack is limited to inferring the bag of words while the order of the words cannot be recovered.
>
> **A3:** Word reconstruction could also pose a privacy risk in instances like exposing trade secrets, healthcare patient diagnosis, or personal identifiable information (PII) such as credit card numbers. Furthermore, consolidated word reconstruction could serve as the first step of sequence reconstruction techniques, followed by iteratively swapping tokens and training them within a language model, as adapted in the prior work FILM and LAMP.
>
> ---
>
> **Q4:** How would the performance change when tying the weights between input embedding and output layer?
>
> **A4:** Our implementation is in the context of weight tying and has demonstrated successful outcomes. We will add such an explanation to our revision.
>
> ---
>
> **Q5:** Will the inference attack still be effective if using secure aggregation as defense? where the server can only observe the aggregated gradients instead of individual’s.
>
> **A5:** Thanks for your question. That is why we need to consider the setting with more training tokens involved. It is worth noting that aggregating (say averaging) gradients from multiple batches (say from various clients) is equivalent to the gradients acquired from the `aggregated' (say concatenating all samples in these batches as a single batch) larger batch.

---

### Official Review · Reviewer_2HVX · 2023-11-01

**Soundness:** 3 good
**Presentation:** 3 good
**Contribution:** 3 good
**Rating:** 6
**Confidence:** 4

**Summary:**

This paper investigates the recovery of the set of words used during federated training of a large language model for the tasks of language modeling and machine translation. The paper proposes an attack, known as “Flat-Chat”, which is able to extract the set of words from the last linear layer’s gradients. To do so, Flat Chat transforms last linear layer gradients and uses a gaussian mixture model to form two clusters (positive/negative), corresponding to tokens which (are/are not) used in the batch, respectively.
The paper also proposes two defenses (freezing and DP-SGD) against the attack.

**Strengths:**

The paper is easy to understand, and the methodology is novel and intuitive. The proposed method demonstrates performant recovery of a majority of tokens from the last linear layer, demonstrating significant leakage of tokens which does not depend on gradients of input embeddings.

**Weaknesses:**

Experimental results could be more comprehensive. In particular, more exploration (e.g. of larger batch sizes) would establish the failure mode of the approach.

I am also curious about the gaussian mixture model of the word types; Does the frequency of each word in the batch impact the quality of the fit? Experiments that demonstrate robustness in this scenario would be helpful in establishing generality of the approach.

Finally, further experiments which show performance of freeze/dp-sgd for language modelling would also help contextualize the benefits and drawbacks of the proposed defenses.

**Questions:**

* I am a bit confused why results for Scratch with the task of Large Language Modelling are not included

* Is the gaussian mixture model accurate at every epoch of fine-tuning? Or is it only at the first epoch?

---

> ### Author Response · Authors · 2023-11-23
> **Response to Reviewer 2HVX**
>
> Thank you for your appreciation and suggestions.
>
> ---
>
> **W1:** more exploration (e.g. of larger batch sizes) on the failure mode of the approach.
>
> **A1:** (1) We have covered some experiments with the *failure* performance of RLG, such as 1,500 and 2,000 tokens in Table 2. (2) When scaling the batch size, the attacker will still be valid as we find **Prec.** score of FlatChat is kept stable or even grows in some cases (see Tables 2 and 3), which benefits from a relatively conservative batch size predictor.
>
> ---
>
> **W2:** Does the frequency of each word in the batch impact the quality of the fit?
>
> **A2:** We consider it as a question out of curiosity instead of an argument of weakness. It is implicitly encoded in our Theorem 3, as scores of high-frequency words will be at the `centre’ of the positive cluster.
>
> ---
>
> **W3:** further experiments which show performance of freeze/dp-sgd for language modelling
>
> **A3:** Yes, we have demonstrated the performance of both freeze and dp-sgd on Machine Translation task in Figure 4.
>
> ---
>
> **Q4:** why results for Scratch with the task of Large Language Modelling are not included
>
> **A4:** In many LLM application scenarios, we directly use well-trained LLM instead of LLM from scratch. Additionally, attacking Scratch is a relatively easier experimental setting for attackers as demonstrated in Machine Translation tasks.
>
> ---
>
> **Q5:** Is the gaussian mixture model accurate at every epoch of fine-tuning? Or is it only at the first epoch?
>
> **A5:** GMM could be easily generalized to various settings:
> 1. The feasibility of the Gaussian Mixture Model is proved by our Theorem 3 based on Theorem 4 (CLT)
> 2. FlatChat empirically works on well-finetuned models, such as MT’s FINETUNE and GPT-2.
> 3. Because the data leakage in the later epochs is of less interest given the data has already been disclosed in the earlier epochs, we focus on the token recovery attack in the early training epoch.
> 4. Any gradient-based reconstruction attacks will fail in an extreme case, say the model overfits a batch of data leading to zero loss and zero gradient vectors.

---

### Meta-Review · Area_Chair_teyD · 2023-12-06

**Metareview:**

This submission suggests a new approach to bag of word recovery in gradient inversion attacks in federated learning. However, the submisson appears unaware of the equivalence of the word recovery problem to the label recovery problem, and thus does not provide comparison to a number of related work and algorithms solving the same task [1,2,3]. The task of word recovery is standard as initial step in recent gradient inversion attacks against language models  [4,5]. The omission of these comparisons makes it hard to contextualize this work, and I encourage the authors to revise their manuscript now that they have been provided additional information on the current state of the field.
Further, it is not always clear how important perfect word recovery really is. In [4,5], sentence recovery can succeed even if word recovery is only inexact, and in sufficiently large batches of data, word frequency can also be estimated from real data and used instead of word recovery.


[1] Yin, Hongxu, et al. "See through gradients: Image batch recovery via gradinversion." Proceedings of the IEEE/CVF Conference on Computer Vision and Pattern Recognition. 2021.
[2] Wainakh, Aidmar, et al. "User-level label leakage from gradients in federated learning." arXiv preprint arXiv:2105.09369 (2021).
[3] Geng, Jiahui, et al. "Towards general deep leakage in federated learning." arXiv preprint arXiv:2110.09074 (2021).
[4] Samyak Gupta, Yangsibo Huang, Zexuan Zhong, Tianyu Gao, Kai Li, and Danqi Chen. 2022. Recovering private text in federated learning of language models. In Advances in Neural Information Processing Systems
[5] Fowl, Liam H., et al. "Decepticons: Corrupted Transformers Breach Privacy in Federated Learning for Language Models." The Eleventh International Conference on Learning Representations. 2022.

**Justification For Why Not Higher Score:**

Missing comparison to related work.

**Justification For Why Not Lower Score:**

N/A

---

### Decision · Program_Chairs · 2024-01-16

Reject